

# Detection and global climatology of two types of cyclone clustering

Chris Weijenborg[1] and Thomas Spengler[2]

[1]Meteorology and Air Quality Section, Wageningen University and Research, Wageningen, the Netherlands
[2]Geophysical Institute, University of Bergen, and Bjerknes Centre for Climate Research, Bergen, Norway

**Correspondence:** Thomas Spengler (thomas.spengler@thomas.spengler@uib.no)

**Abstract.** Cyclone clustering, the swift succession of multiple extratropical cyclones during a short period of time, is often associated with weather extremes and characterised by a strong atmospheric jet and enhanced baroclinicity. While several diagnostics exist to detect cyclone clustering, most focus on a regional impact. We introduce a novel global detection for cyclone clustering, inspired by the original idea of cyclone families by Bjerknes and Solberg, in which individual cyclones

follow a similar track. We further subdivide cyclone clusters into two types, a 'Bjerknes' type and a stagnant type. The former is associated with cyclones that follow each other over a minimum distance, whereas the stagnant type requires a proximity over time while these cyclones do not move much in space.

We find that cyclone clustering is most frequent along the storm tracks, with more cyclone clustering during winter compared to summer. The majority of cyclone clustering occurs just south of the main storm tracks in the Atlantic and Pacific basins.

In the Southern Hemisphere, most cyclone clustering is found in the South-Indian Ocean. Bjerknes type cyclone clustering is associated with stronger cyclones compared to non-clustered cyclones, while for the stagnant type this intensity difference is less pronounced. This effect is strongest for the North Atlantic and North Pacific, while clustered cyclones in the South Indian Ocean are generally not much stronger. The cyclone intensity within the Bjerknes type does not decrease during a cluster, while in contrast secondary cyclones of the stagnant type are significantly weaker than primary cyclones. This suggests that these

two types of cyclone clustering are dynamically different.

## 1 Introduction

Cyclone clustering, the rapid succession of extratropical cyclones during a short period of time, is often associated with European weather extremes, such as extensive wet spells (Moore et al., 2021) and strong wind gusts yielding large economic losses (Priestley et al., 2018). The idea that several cyclones follow a similar track dates back to the concept of cyclone families by

Bjerknes and Solberg (1922). To investigate the dynamical causes of cyclone clustering, it is desirable to detect and characterise the occurrence of cyclone clustering. So far, focus has either been on local impact-based detection or on a statistical analysis of storm recurrence (e.g. Mailier et al., 2006; Vitolo et al., 2009; Priestley et al., 2017). While the former cannot be applied globally, the latter is difficult to relate to individual clustering events. We therefore introduce a novel global cyclone cluster detection that is closer to the original concept of cyclone families by Bjerknes and Solberg (1922).

Based on the idea that cyclones occur more regularly over time in some regions, Mailier et al. (2006) defined cyclone clustering (serial clustering in their paper) using a dispersion diagnostic, comparing local occurrence of cyclones with a random



Poisson process. They refer to a region as underdispersive when the monthly cyclone occurrence at a particular location is more regular than expected from a Poisson process. In contrast, a region is overdispersive when cyclones occur less regularly compared to a Poisson process. The latter is associated with cyclones clumping together in time as clusters and is mainly found

at the exit regions of the North Atlantic and North Pacific storm tracks. Similar algorithms have been applied by Kvamstø et al. (2008); Vitolo et al. (2009); Pinto et al. (2013); Economou et al. (2015). Future changes in this dispersion statistic are generally small (Economou et al., 2015). A problem with this statistical definition, however, is that it defines clustering in a relative sense. The diagnostic does neither quantify how many cyclone clusters pass at a particular location nor identify which cyclones are part of a particular cluster.

Another set of diagnostics for cyclone clustering counts the number of cyclones at a particular location during a short period of time (Priestley et al., 2017; Bevacqua et al., 2020). For example, Priestley et al. (2017) defined clustering off the coast of western Europe as the occurrence of at least 4 cyclones in a period of seven days within a radius of 700 km around that location. Using composites of clustered events in this way, they found that clustering at the storm track exit is related to a strong extended jet, flanked by double sided Rossby wave breaking. A similar algorithm was used by Bevacqua et al. (2020),

but using a maximum temporal distance between cyclones of one day, instead of counting cyclones in the seven day running mean. While these algorithms distinguish which cyclones are clustered, they still rely on a local impact-based definition of cyclone clustering.

Priestley et al. (2020b) extended the method of Priestley et al. (2017) to distinguish if detected cyclones form along the trailing cold front of a previous cyclone. This allows to distinguish between primary and secondary cyclones, which is a useful

classification as clustering is often associated with secondary cyclogenesis (Pinto et al., 2014). Priestley et al. (2020b) found that about 50% of the cyclones are clustered along the Atlantic storm tracks. Although this algorithm is less local than the previous algorithms, it relies on both a frontal as well as a cyclone detection. Detecting fronts relies on several choices and is thus sensitive to the chosen variable for detection (Thomas and Schultz, 2019). Furthermore, this algorithm a priori assumes that clustered cyclones are always due to secondary cyclogenesis.

There have also been attempts to investigate the similarity of tracks of extratropical cyclones, for example Blender et al. (1997) used k-means clustering based on the cyclone displacement relative to its genesis location. They divided North Atlantic cyclone tracks into zonal, north-east moving, and stationary types. This definition of clustering ensures that different cyclones follow tracks in the same direction, though it does not take the temporal component into account. Therefore, this definition puts all cyclones travelling in a zonal direction in the same cluster, independent if they occur shortly after each other or not. This

might not be desirable, especially if one is interested in potential dynamical differences between different types of clusters.

The diagnostics outlined above either only have a local criteria for proximity of tracks or put all cyclones moving in a similar direction into one cluster. However, to disentangle the mechanisms of cyclone clustering, one needs an algorithm closer to the original idea of cyclone clustering of (Bjerknes and Solberg, 1922). Ideally, a clustering diagnostic should:

– check if two tracks are close in space-time over a considerable amount of distance and/or time

– detect which cyclones are members of a specific cluster





- be unbiased with respect to the clustering mechanism

- be applicable globally

Synoptically, clustering over the Atlantic is often associated with strong, elongated jets and secondary cyclogenesis along trailing cold fronts of preceding cyclones (Pinto et al., 2014; Priestley et al., 2017; Weijenborg and Spengler, 2020). Stronger

jets correspond to higher baroclinicity, which explains that clustered storms tend to be more intense (Vitolo et al., 2009). However, given that several cyclones follow a similar track, one needs to explain how this baroclinicity is maintained. Given the importance of latent heating for the maintenance of baroclinicity (Hoskins and Valdes, 1990; Papritz and Spengler, 2015), Weijenborg and Spengler (2020) proposed that cyclone clustering could be caused by strong latent heating along trailing cold fronts.

As the focus has mainly been on clustering in the North Atlantic, with articles on cyclone clustering in the Southern Hemisphere being particularly sparse, we propose a new algorithm that can be applied globally. The algorithm defines cyclone clusters based on individual cyclones following a similar track for a certain length or time. We also introduce two types of clusters, one resembling the original idea of Bjerknes and Solberg (1922) with cyclones following a similar track over a long distance and a stagnant type, where cyclones do not move much over their life time and therefore resemble clusters at the

storm tracks exit. We present a global climatology of both types and discuss differences in intensity between clustered and non-clustered cyclones as well as differences between cyclone intensity within a cluster.

## 2 Data and Methodology

We use the ERA-Interim reanalysis from the European Centre for Medium Range Weather Forecasts (ECMWF) (Dee et al., 2011), which is available at a triangular truncation of T255 with a 6-hourly time interval providing analyses at 00, 06, 12, and

18 UTC. We interpolated the data onto a 0.5-degree grid and use the mean sea level pressure to track extratropical cyclones.

We use the University of Melbourne cyclone detection and tracking algorithm (Murray and Simmonds, 1991a, b). The algorithm detects cyclones as maxima in the Laplacian of the mean sea level pressure and tracks them over time using a nearest-neighbourhood method together with the most probable direction of propagation (Murray and Simmonds, 1991a, b; Michel et al., 2018). We use the same parameters as in Tsopouridis et al. (2021) and select cyclone tracks that last at least 24

hours. However, in contrast to Tsopouridis et al. (2021), we do not pre-select any threshold on storm intensity and do not apply any requirements on a minimum distance travelled by cyclones. We decided against these additional criteria, because we want to investigate if clustered cyclones are stronger compare to non-clustered cyclones. We do not include a distance criterion to detect all cyclones belonging to the stagnant type. To minimize the influence of orography, we discard cyclones located above 1000 meter.





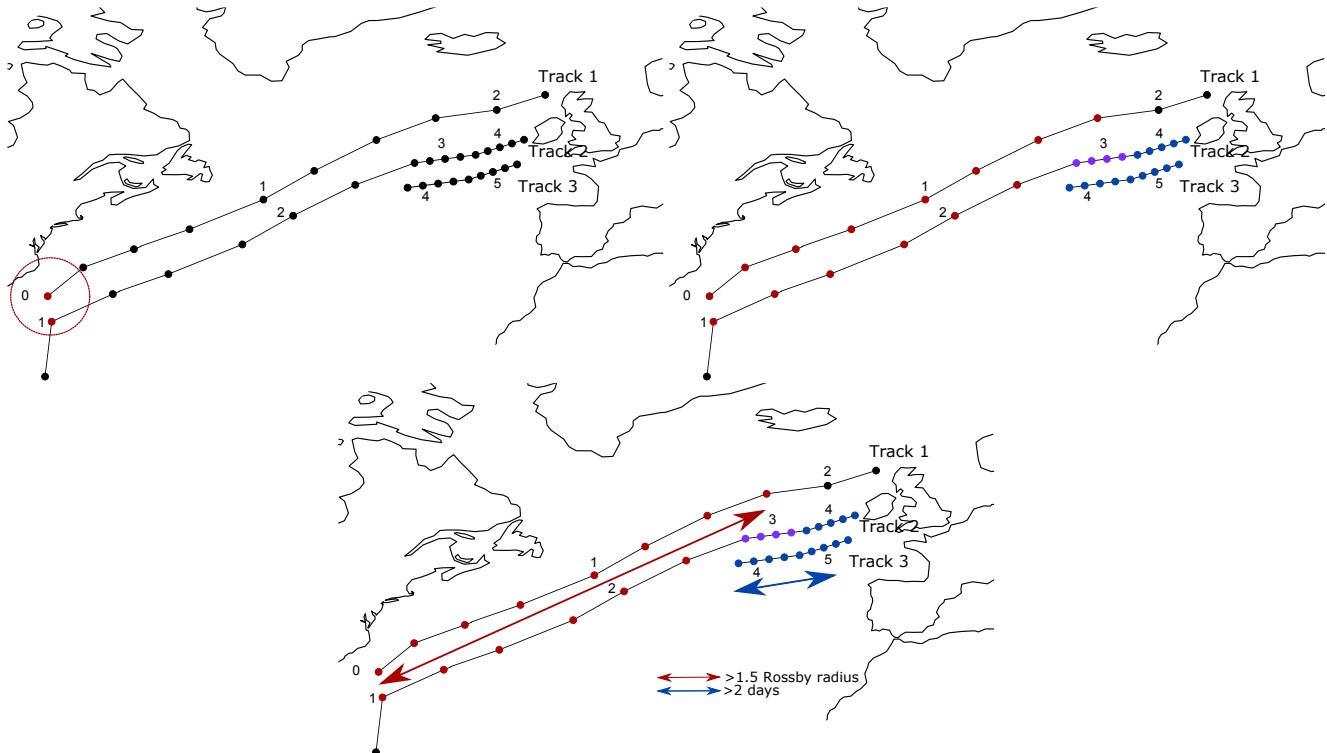

**Figure 1.** Schematic of cyclone clustering detection. Black lines indicate three different cyclone tracks. Time in days since track 1 started is indicated by the timestamp next to each point. (a) Red radius in indicates the distance threshold (Rossby radius of deformation) for one example point along track 1. (b) All points connected according to the criteria for the local space-time proximity are indicated by coloured points, with red points indicating a connection between track 1 and 2, and blue points indicating a connection between track 2 and 3, and purple points along track 2 are connected to both track 1 and 2. (c) Indication of the overlap in space (red arrow) and time (blue arrow) used in the second step of the algorithm. In this example, track 1 and 2 satisfy the length overlap criterion, while tracks 2 and 3 satisfy the time overlap criterion.

## 2.1 Cyclone cluster detection

Conceptionally, clustered cyclones follow each other for a significant distance or time. Hence, for every pair of cyclone tracks we first check if they are close enough to each other in space and time (See Figure 1). We check pairwise points along two tracks if the spatial distance is within $\delta x_{local}$ of one Rossby Radius of deformation ($L_R = NH/f_0$) and within a temporal period $\delta t_{local}$ of 36 hours (indicated by the red dots in Figure 1a). The Rossby radius of deformation is a measure of the wavelength of maximum growth for baroclinic instability (Holton, 2004) and therefore sets the typical size of an extratropical cyclone. The choice of 1.5 days is roughly the median time passed between the occurrence of mid-latitude cyclones in the North Atlantic and North Pacific storm track regions in winter (not shown).



The approach described in Figure 1a is very similar to other approaches detecting cyclone clustering. The main difference is that instead of only checking for local proximity of two cyclone tracks, we check for every pair of cyclone track points along two tracks if they are close together in space-time, indicated by all the coloured dots in Figure 1b.

In the second step, we assess the overlap between the two cyclones travelling along a similar track in distance $\delta x_{overlap}$ and/or time $\delta t_{overlap}$ (coloured dots in Figure 1b). For all the connected points in step 1, we check if the maximum distance between them is either larger than 1.5 Rossby radius (measured by the great circle distance between the first and last red dot in Figure 1c) or that the temporal difference between them is more than 2 days (the time elapsed between the first and last red dot in Figure 1c). If either one of the two conditions is satisfied, the two cyclone tracks are connected as a cluster. The final step is combining these uniquely connected cyclone tracks, where each cyclone track in a cluster is connected to at least one other cyclone track, but not to any other cyclones outside the cluster (see Figure 1c).

We choose a length overlap $\delta x_{overlap}$ of at least 1.5 Rossby radius to have a minimum length of overlap significantly longer than the typical size of a cyclone. The time overlap $\delta t_{overlap}$ of 2 days comprises a significant part of the cyclone lifetime. We performed sensitivity tests on both the time and length overlap. One could also have chosen slightly less strict thresholds, e.g. 1 Rossby radius and 1.5 days. However, while not qualitatively altering the results, these choices would lead to extremely long clusters, especially in the Southern Hemisphere (not shown). A further argument to choose the more strict parameters is to prevent that cyclones from different clusters end up in the same cluster.

The above method yields all cyclone clusters, regardless if cyclones follow each other over a long distance or an extensive period of time. However, as the two types of clusters might be dynamically different, we distinguish between them and present climatologies for each category. We refer to these as the Bjerknes type and stagnant type, dependent on if they fulfill the length or time criterion, respectively. We explicitly exclude the length criterion for the stagnant type, because they should represent clusters that do not move much in space. For the schematic example in Figure 1c, tracks 1 and 2 form a Bjerknes type cluster, while tracks 2 and 3 are part of a stagnant cluster.

The Bjerknes type represents cyclone families described by Bjerknes and Solberg (1922), while the stagnant type represents clusters of cyclones that do not travel far. As an individual cyclone can be simultaneously part of a Bjerknes and a stagnant type cluster, there is a chance for double counting cyclones. For example, in Figure 1 cyclone 2 is part of both a Bjerknes as well a stagnant cluster. Hence, the cyclone track densities of Bjerknes and stagnant type clustered cyclones are not additive and the sum can thus be larger than the density of all clustered cyclones.

As in Priestley et al. (2020b), we define cyclones that are not part of a cluster as 'solo' cyclones. We compare both differences in location as well intensity between solo and clustered cyclones. To distinguish between primary and secondary cyclones, cyclones are ordered by the first time step they are connected with any cyclone in that cluster (coloured dots in 1). This time step might be different than the genesis location. For example, for track 2 in the example in Figure 1, the first time step is not 'clustered'. Note that for clustered cyclones, only the connected parts are used (coloured dots in 1). Therefore, the fractional densities of solo and clustered cyclones do not add to 100%.

To test the hypothesis that clustered cyclones are stronger, we use the maximum Laplacian in a small 1.25 degree radius around the centre during the lifetime of a cyclone to define the cyclone strength (similar as in Tsopouridis et al. (2021) and





Michel et al. (2018)). As the geostrophic relative vorticity is inversely proportional to the Coriolis parameter and therefore the latitude $\phi$, we normalise the Laplacian with $sin(\phi)$.

We choose the maximum of the normalised Laplacian instead of minimum pressure, as it is directly related to the geostrophic vorticity and thus the strength of the wind speed associated with a cyclone. Qualitatively, however, the results are not very sensitive to this choice, given that cyclones with a larger Laplacian are commonly associated with a deeper minimum in pressure. Furthermore, we define the intensity of a cluster using the maximum normalised Laplacian of the strongest cyclone in a cluster. The results are similar when choosing the mean intensity of cyclones in a cluster (not shown).

## 140 3 Climatology of clustered cyclone tracks

### 3.1 Winter

The occurrence of clustering in winter generally aligns with the climatological storm tracks for both the North Atlantic as well the North Pacific with clustered cyclones occurring about 8-10% (Figure 2b). In contrast, very few clustered cyclones are found in the Mediterranean and Barents Sea. While this is similar to Priestley et al. (2020a), though with slightly lower absolute

values, our findings are in contrast to Mailier et al. (2006), who detected serial clustering mostly at the storm track exits. This difference is mainly due to our diagnostic determining absolute number of clustered storms, which is highest along the storm tracks. The dispersion diagnostic from Mailier et al. (2006), on the other hand, determines irregularities in the occurrence of cyclones in a given month, which is highest at the storm track exit where the variability of the location of the jet is largest (Woollings et al., 2010).

In contrast to the clustered cyclones, solo cyclones occur more regularly at the storm track exit (Figure 2a). Moreover, there are several additional regions where solo cyclones occur more regularly in the North Atlantic: in the lee of the Rocky mountains and over the Mediterranean sea. Priestley et al. (2020b) identified similar regions with high solo cyclone density, though with less solo cyclones around the Norwegian coast. Reason for this difference is most likely that they detect much fewer cyclones in this region in general. In the Pacific basin, solo cyclones occur more often at the storm track exit.

The fraction of clustered cyclones in the Northern Hemisphere is about 40 to 50 % of the total number of storms (see Figure 2 c). Priestley et al. (2020b) found even larger fractional track densities of up to 60 % over North Atlantic. In the North Pacific, the fractional density of clustered storms is slightly higher than in the North Atlantic. Highest fractional densities are found just south of the main storm tracks in both the North Atlantic and North Pacific (see Figure 2c). Moreover, the fractional densities are oriented less northward compared to the climatological storm tracks, especially in the North Atlantic. This indicates that

clustering occurs more often for more zonally oriented storm tracks, which is also the case for cyclone clustering defined as cyclones associated with secondary cyclogenesis (Priestley et al., 2020b). However, we also find large clustering frequencies along the Norwegian coast at the storm track exit in the Atlantic.

There are two main genesis regions in the North Atlantic for clustered cyclones, firstly near the Gulf Stream and secondly in an area near Greenland (not shown). These genesis regions are partly similar to Priestley et al. (2020b), who found that

cyclones forming due to secondary cyclogenesis mainly have genesis in these regions. In the North Pacific, genesis occurs





**Figure 2.** Climatology of cyclone clustering during the winter seasons. (a) Density of solo cyclones (not clustered) in a 1000 km$^2$ area for DJF in the Northern Hemisphere. (b) As (a), but for clustered cyclones. (c) Fractional density of clustered cyclones (shading). (d-f) as (a-c), but for the Southern Hemisphere for JJA. Black boxes in (b) and (e) indicate chosen regions in section 4, and black contours in each panel indicate climatological storm tracks of all cyclones (contours at 10, 15 & 25 % per 1000km$^2$).

generally more on the western side of the basin over the Kurushio region (not shown). This indicates that clustered cyclones travel over the entire basin in the North Atlantic and North Pacific.

For the Southern Hemisphere winter season, cyclone cluster density is highest in a small band around Antarctica, with the highest densities over the South Indian Ocean (Figure 2e). Absolute numbers of clustering are higher compared to the Northern

Hemisphere, which is partially due to higher cyclone densities in general. The fraction of clustered cyclones is about 35-40 %, which is comparable to the Northern Hemisphere (Figure 2f). Furthermore, the genesis region is less clear compared to the Northern Hemisphere, with genesis of clustering mainly occurring in the same band as the storm tracks (not shown).





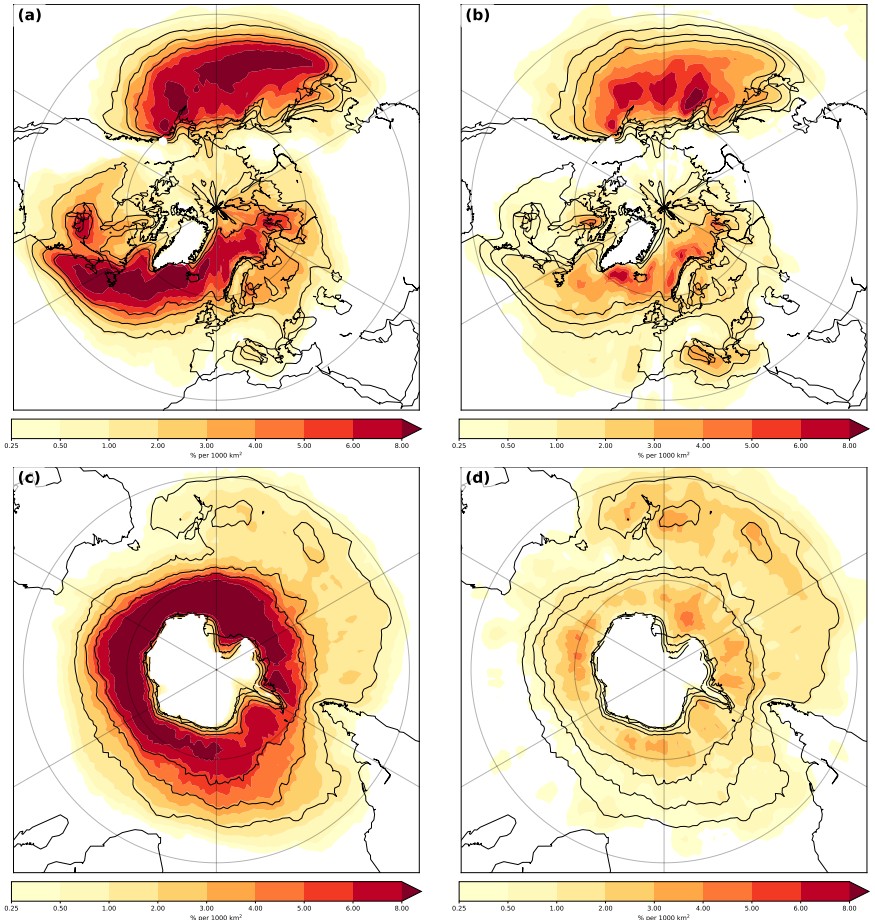

**Figure 3.** Densities for (left) Bjerknes type cyclone clusters and (right) stagnant type for (top) Northern Hemisphere and (bottom) Southern Hemisphere during the winter season. Shading denotes fraction of times of a clustered cyclone at a location in a 1000 km$^2$ area. Black contours indicate clustered densities (at 2, 4 & 6 % per 1000km$^2$)

The Bjerknes-type clusters occur all over the North Atlantic and North Pacific regions, but relatively more at the entrance and just south of the storm tracks (Figure 3a). In contrast, the stagnant type occurs more at the storm track exit in the North Atlantic

and North Pacific as well as more to the north of the main storm tracks compared to the Bjerknes type (Figure 3b). Moreover, we detect more cyclones of this type in the North Pacific. In contrast, for the Southern Hemisphere, absolute numbers of (clustered) cyclones of the stagnant type are small (Figure 3d). While Bjerknes type clustered cyclones frequencies are about 6-8 % along the storm tracks (Figure 3c), stagnant clustered cyclones frequencies are only 1 to 2 %. Fractional densities for the stagnant type are lower, with more than 10% clustered stagnant cyclones around Australia, east of Australia, and near

Madagascar. In contrast, the fractional Bjerknes clustered densities are highest just equatorward of the main storm tracks.







**Figure 4.** As Figure 2, but for the respective summer seasons.

## 3.2 Summer

For summer, the frequency of clustered storms in the Northern Hemisphere is significantly reduced and shifted to the western side of the basin, consistent with weaker storm tracks during summer (Figure 4b). While Mesquita et al. (2008) found a northward shift of cyclones, especially on the western side of the basins, this is not evident for clustered cyclones in the Northern Hemisphere (Figure 4b and c). Clustered cyclones in summer occur less often at the storm track exit in the Northern Hemisphere. Genesis is also slightly shifted to the west with less genesis of clustered cyclones in the lee of Greenland (not shown). For the Southern Hemisphere, there are no larger differences in the occurrence of clustered cyclones between summer and winter (Figure 4e-f). Solo cyclones in summer occur mainly within the storm tracks, suggesting a westward shift compared to winter (Figure 4a). The shift to the west is less clear in the Southern Hemisphere (Figure 4d-f), but densities for clustered cyclones are higher in the South Atlantic compared to further east in the South Indian Ocean.





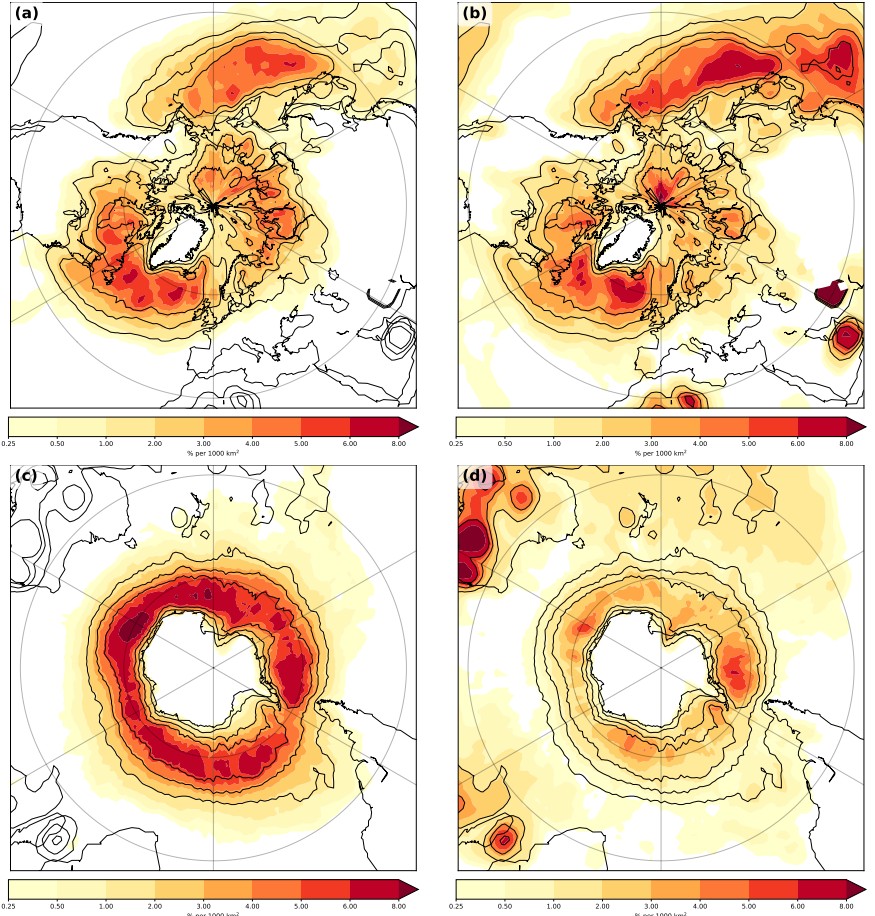

**Figure 5.** As Figure 3, but for the respective summer seasons.

The different cluster types are reduced in summer, with a stronger reduction in the Bjerknes-type (Figure 5a-b). This is intuitive, as the jet strength is significantly reduced in summer, which mainly appears to affect the frequency of the Bjerknes-type clusters (Figure 5a). Furthermore, there is a shift towards the western side of the basins in the North Atlantic and North Pacific. However, for the stagnant clusters, cyclone densities are similar to winter. For the Southern Hemisphere, there is a decrease in the Bjerknes-type clusters and a small increase in stagnant clusters (Figure 5c-d).

## 4 Characteristics of clustered cyclones

Some studies argue for a systematic mechanism associated with cyclone clustering Priestley et al. (2017); Weijenborg and Spengler (2020) and that clustering is generally associated with stronger cyclones Vitolo et al. (2009). We test these findings by assessing differences in length and storm intensity, both for all clusters as well for the two sub-types of clustering. In this





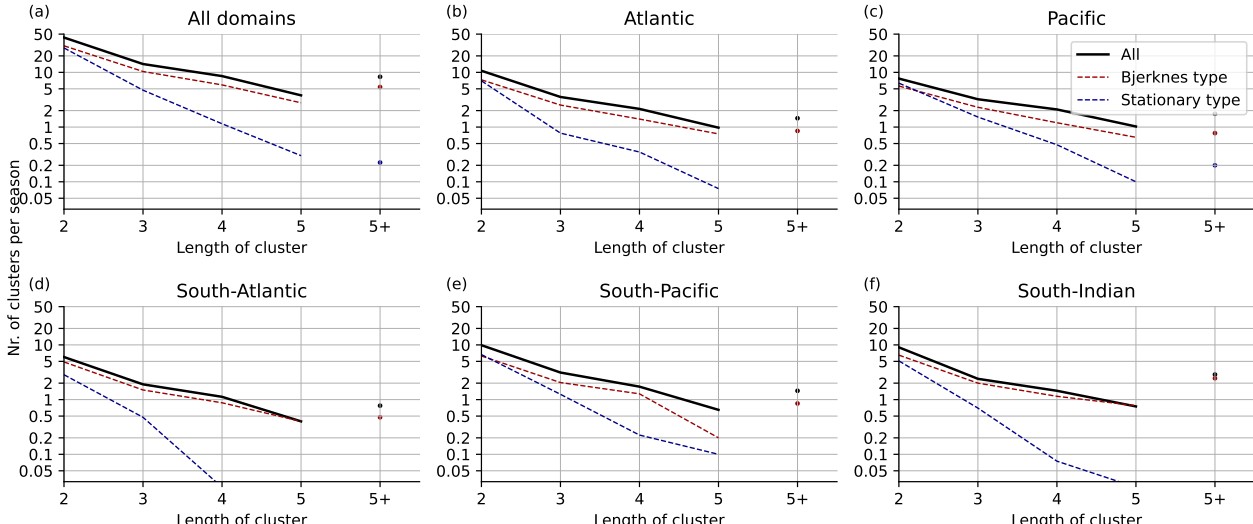

**Figure 6.** Number of clusters for the respective winter season as function of cluster length for all clusters (black line), Bjerknes type (red line), and stagnant type (blue line). (a) Top left for all regions. (b-f) Other panels for Atlantic, Pacific, South Atlantic, South Pacific, and South Indian Ocean, respectively.

section, we only investigate cyclones during the winter season (December until February for the Northern Hemisphere, and June to July for the Southern Hemisphere), as extratropical cyclones have the highest occurrence and intensity during winter.

We define the length of a cluster as the number of cyclones in a cluster and use the cyclones occurring in the regions introduced in Figures 2b and d. In general, the likelihood of having a cluster of length $n$ decays exponentially (Figure 6a). This decay is stronger for stagnant clusters. While there are no big differences between the different basins, there are relatively more

longer stagnant clusters for the North Atlantic and North Pacific (Figure 6b-c) compared to the Southern Hemisphere (Figure 6d-f). In contrast, specifically for the South Atlantic and South Indian Ocean, there are relatively few stagnant clusters (Figure 6e).

## 4.1 Cyclone intensity

Comparing the intensity of the strongest cyclone per cluster with the strength of the strongest cyclones in a randomly chosen

selection with the same number of cyclones as in the cluster, we find that clustered cyclones are generally stronger and that solo cyclones are generally weaker (Figure 7). The qualitative differences between the different basins are small with slightly stronger clustered cyclones in the North Atlantic and North Pacific. The differences in intensity for clustered and non-clustered cyclones is, however, less in the three basins in the Southern Hemisphere. Specifically the South Indian Ocean and South Pacific stand out with only a small difference in intensity. This indicates that clustering might be dynamically different for the

storm tracks in the Northern and Southern Hemisphere.




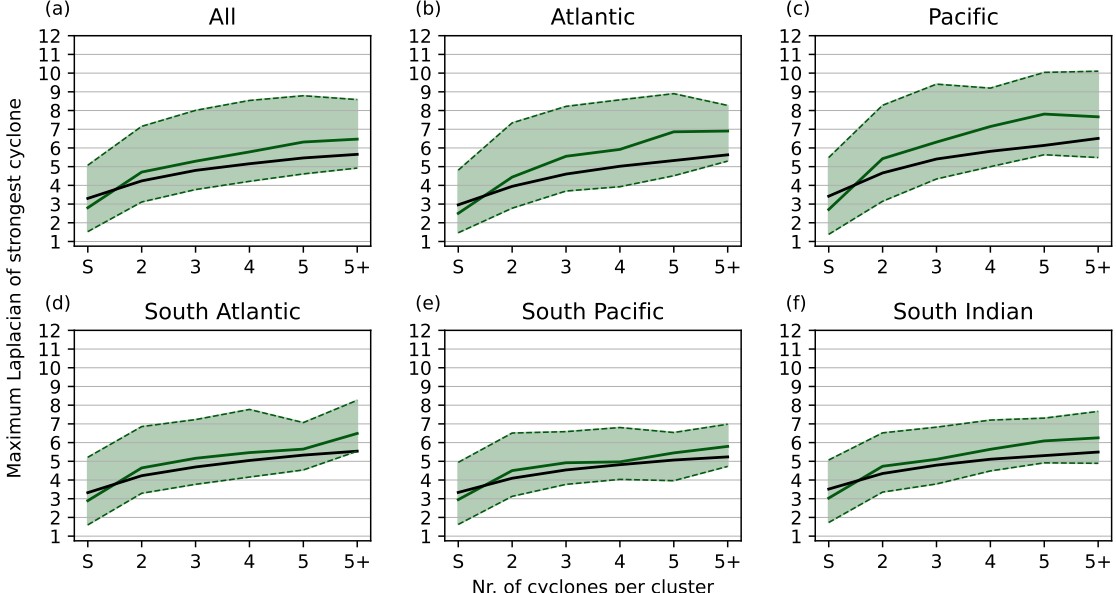

**Figure 7.** Cyclone intensity as function of cluster length, i.e. the number of storms in a cluster. The bin denoted with "S" indicates the strength of solo (non-clustered) cyclones. Green solid line indicates median values and variability between the 10 and 90 % quantiles is indicated by shading. The black line indicates expected value from randomly chosen clusters.

The strongest cyclones in Bjerknes type clusters are stronger compared to randomly selected cyclones (Figure 8a). One might have anticipated this result for Bjerknes type clustered cyclones as they are associated with a stronger jet and baroclinicity. This difference is largest for the North Atlantic and the North Pacific (Figure 8b-c), while there are only small differences for the South Indian Ocean.

In contrast, the median of the strongest cyclones in stagnant clusters of length $n$ falls between that of Bjerknes clusters and the expected value of a randomly chosen cyclone (Figure 8a). The 90% quantile for the stagnant type is also lower as that of the Bjerknes type, with the difference in intensity being stronger for the North Pacific compared to the North Atlantic (Figure 8a and b). For the Southern Hemisphere the intensity between stagnant clustered cyclones and solo cyclones is very similar(Figure 8d-f). This difference in intensity between the two types of clusters suggests that they are dynamically different.

To investigate the local impact of clustered cyclones, we determine how often a cyclone is present with an intensity higher than the 90 % quantile of the intensity at that particular location. We do this for both clustered and non-clustered cyclones. Even though the number of clustered cyclones is lower, the absolute number of intense clustered cyclones is higher than that of intense non-clustered cyclones (Figure 9a and b). This is especially the case along the storm tracks in the North Atlantic and North Pacific as well as just north of the United Kingdom and along the coast of Norway. For the Pacific storm track exit this is less clear, with even higher densities of intense non-clustered cyclones along the coast of the United States. However, for both



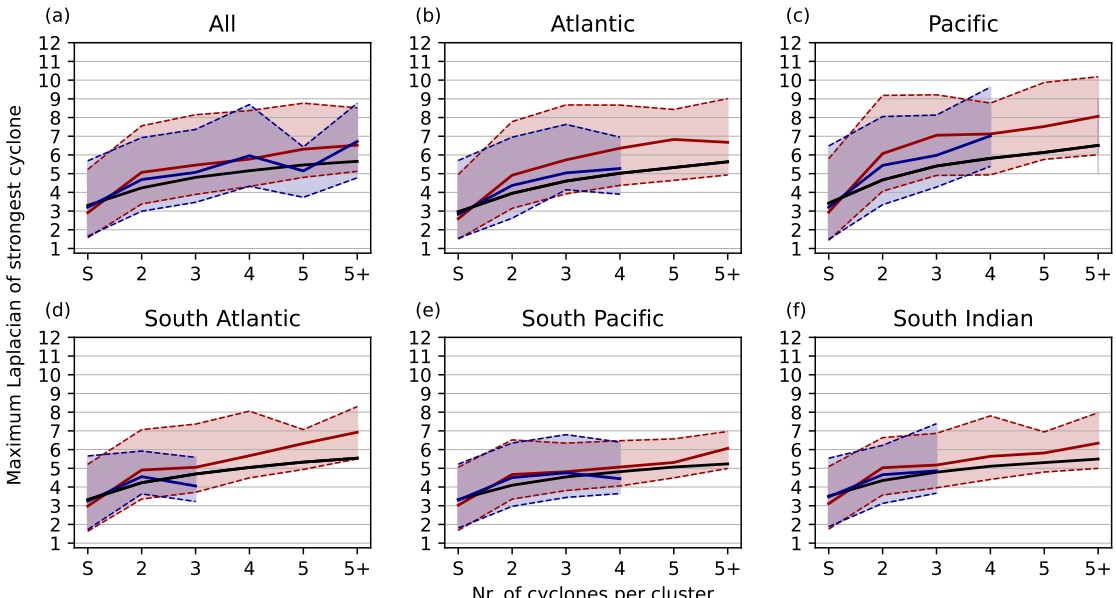

**Figure 8.** As Figure 7 but for Bjerknes type (red solid line and shading, for mean and 10 to 90 % quantiles) and stagnant type (green shading). Black line indicates expected value from randomly chosen clusters.

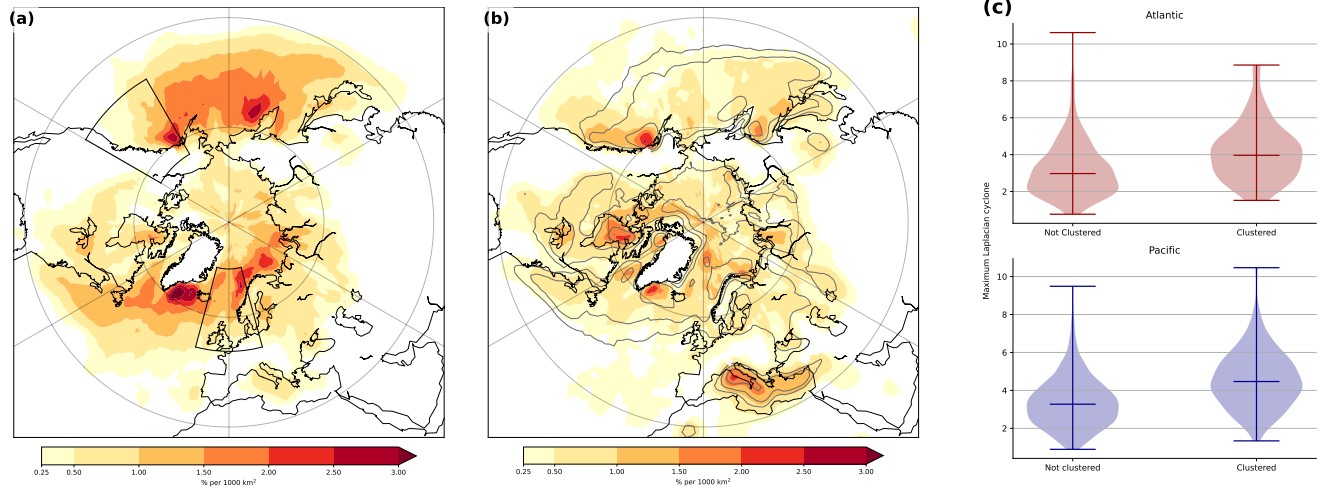

**Figure 9.** (a) Density of clustered cyclones with an intensity of at least the 90 % quantile at that location during the winter season. (b) as (a), but for non-clustered cyclones. (c) Violin plot of intensity of clustered and non-clustered cyclones at the storm track exit regions indicated by the black boxes in (a).





the Atlantic as well the Pacific storm track exit regions the intensity of clustered cyclones is shifted towards higher intensities (Figure 9c).

## 4.2 Cluster length and cyclone intensity

Given that Bjerknes type clusters are associated with a strong baroclinicity and jet, we investigate the relation between the

strength of cyclones and the length of overlap $\delta x_{overlap}$ along their tracks in space-time. If a cyclone is connected to multiple cyclones, the maximum overlap $\delta x_{overlap}$ is used. This maximum overlap is a measure on how 'clustered' a specific cyclone is. Solo (non-clustered) cyclones are put in the lowest bins ($\delta x_{overlap} < 1.5$ Rossby Radius in Figure (a) and $\delta t_{overlap} < 2$ days in (b)).

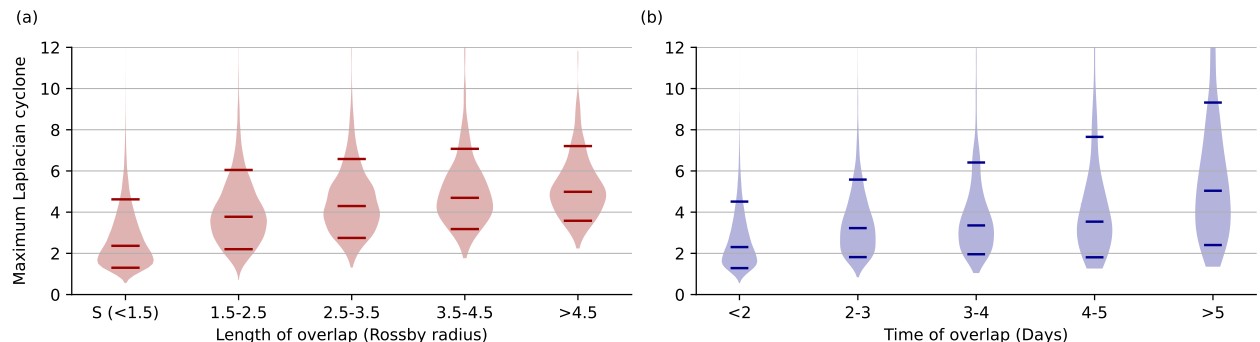

**Figure 10.** Violin plots for cyclone intensity for (a) Bjerknes type clusters as function of length as function of Rossby radius and (b) Stagnant type clusters as function of time overlap. The bin denoted with "S" indicates the intensity of solo (non-clustered) cyclones. The Medians and 10and 90 % quantiles in each violin plot are indicated by solid lines.

There is an increase in cyclone intensity, with respect to the length of overlap (Figure 10a), especially up to about three

Rossby radius. This is consistent with previous studies suggesting that intense cyclones travel over a large distance (Wang and Rogers, 2002). Specifically over the Gulf stream and Kuroshio one can expect more intense and bomb cyclones Sanders and Gyakum (1980); Wang and Rogers (2002), which are likely associated with cyclone clustering.

Contrary to the Bjerknes type clusters, cyclones clustered according to the stagnant type feature a weaker increase in intensity during the lifetime of a cluster (Figure 10b). The median for the stagnant type clusters increases up to 50 % compared to solo

cyclones, while the median for cyclones of the Bjerknes type clusters increases up to almost twice that compared to solo cyclones. This again indicates that the two types of clusters are dynamically different.

## 4.3 Cyclone intensity within a cluster

We showed that clustered cyclones are more intense than solo cyclones. To check if there are consistent differences in cyclone intensity within a cluster, we select clusters of at least length $n = 3$ and distinguish between 'primary' (first), secondary+, and





final cyclones in a cluster. There are only small intensity differences in intensity within Bjerknes type clusters (Figure 11a), suggesting the existence of processes that replenish baroclinicity during the clustering period, as suggested by Weijenborg and Spengler (2020). The last cyclone in a Bjerknes type cluster is slightly less intense than the previous cyclones in the cluster.

In contrast, there is a decrease in cyclone intensity during the lifetime of stagnant cluster with the primary cyclone being the strongest (Figure 11b). This primary cyclone is almost as strong as the primary cyclone in Bjerknes type clusters, though subsequent cyclones are less intense, with the final cyclone being the weakest. This decrease in intensity is both visible in the median as well as in the 10 and 90 % quantiles.

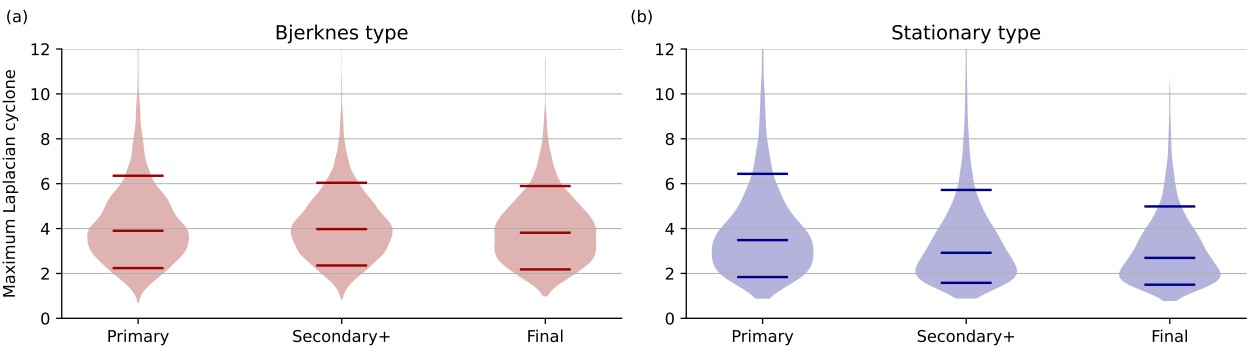

**Figure 11.** Violin plots for the intensity of cyclones for all clusters (left), Bjerknes type (middle), and stagnant type (right) for the first (Primary), all secondary, and final cyclones in each cluster. Medians and 10% and 90% quantiles are indicated by horizontal lines.

## 5 Conclusions

Most existing cyclone clustering diagnostics focus mainly on impact and use local measures, in contrast to the original idea by Bjerknes and Solberg (1922) that cyclone families form along the polar front and following a similar track. Therefore, we introduce a novel cyclone cluster diagnostic that can be used globally. Our clustering diagnostic identifies if multiple cyclone tracks are in close proximity in space and time. We subdivide cyclone clusters into two different sub-types, which we refer to as Bjerknes and stagnant types, where cyclones in the former category need to travel over a minimum distance along a similar track, whereas the latter contains less mobile cyclones occurring in a similar region over a given time.

Using our diagnostic, we find that cyclone clustering mainly occurs near the main storm tracks in the North Atlantic and North Pacific, with the highest fraction of clustered cyclones just to the south of the storm tracks. In the Southern Hemisphere, highest frequencies are found in the South Indian Ocean. In general the Bjerknes-type cluster is found more towards the storm track entrance, while stagnant clusters are more frequent at the storm track exit.

Clustered cyclones are stronger on average than non-clustered cyclones, with this difference increasing with the number of cyclones in a cluster. This increase in intensity is stronger for Bjerknes type cyclones, for which the intensity of cyclones also

increases when the distance that cyclones follow each-other increases. This suggests a replenishment of baroclinicity, which is likely related to diabatically induced secondary cyclogenesis (Weijenborg and Spengler, 2020). In contrast, for the stagnant type, cyclones are not stronger compared to non-clustered cyclones, suggesting that the mechanisms generating the clusters are different for the two types. There are some regional differences between the Northern and Southern Hemisphere, with generally stronger clustered cyclones in the Northern Hemisphere.

Our results are consistent with previously published climatologies of cyclone clusters (Mailier et al., 2006; Priestley et al., 2020b). As in Priestley et al. (2020b), clustering in the Northern Hemisphere winter mainly follows the storm track, with genesis occurring often at the storm track entrance at the Gulf stream and the Kuroshio regions. One difference between our results and that of Priestley et al. (2020b) is that we did not find a shift in genesis between the first and secondary cyclones in a cluster. This could be due to the differences in detecting cyclone clusters, as Priestley et al. (2020b) explicitly demands that

secondary cyclones have their genesis along a trailing cold front of a primary cyclone. Moreover, our results are different to Mailier et al. (2006), who found the highest frequency of clustering at the storm track exit. This differences can be attributed to the statistical nature of their algorithm. While their algorithm focuses on the regularity of cyclone occurrence in a given month, our algorithm determines the absolute number of clustered cyclones.

Based on the global applicability of our novel detection and classification, future research can address underlying mecha-

nisms of cyclone clustering and investigate regional differences. Furthermore, due to our sub-categorisation into two different types of cyclone clustering, one can assess dynamical differences in the initiation and evolution of Bjerknes and stagnant type of cyclone clusters. Last but not least, given that our algorithm also distinguishes between primary and secondary cyclones in a cluster, one can further differentiate how cyclones within a cluster influence each other. The latter is of particular interest when considering the mechanism of secondary cyclogenesis and maintenance of baroclinicty during Bjerknes type clusters.

*Author contributions.* CW performed the data analyses and prepared the figures. TS contributed to the interpretation of the results and to the writing of the paper.

*Code and data availability.* The ERA-Interim reanalysis (Dee et al., 2011) used in this study is publicly available. The cyclone detection algorithm is available as part of *dynlib*, a library of meteorological analysis tools (Spensberger, 2024).

*Competing interests.* The authors declare that they have no conflict of interest.

*Acknowledgements.* The authors thank the European centre of median range weather forecast to make the ERA-Interim data set openly available. Python library *Dynlib* (Spensberger, 2021). This study was supported by the Research Council of Norway (Norges Forskningsråd, NFR) through the BALMCAST project (NFR grant number 324081).



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
