# Peer review of "Detection and global climatology of two types of cyclone clustering"

_EGUsphere, 2024_

## Referee Comment (RC1)

**Review of *'Detection and global climatology of two types of cyclone clustering'* by Weijenborg and Spengler**

**Summary**
This work introduces a new climatology of cyclone clustering using a novel detection and classification algorithm. The work and associated findings are interesting, and some new insights are provided, however I question the definition of clustering used here compared to some of the previous scientific literature. The authors use a fixed clustering threshold everywhere and largely identify clustering in the core of the storm track, where more cyclones are found. Whereas previous efforts have identified clustering as abnormal periods of high cyclone activity. Therefore, I would like to see more justification from the authors as to their methodological choices and explanation of novelties relative to prior studies, and how this work differs from a simple classification of the storm tracks. I recommend major revisions for this work, and detail my points, both major and minor, below.

**Major Comments**
1. My main concern surrounds the choice of the algorithm and justifications made by the authors. The method is to group cyclone travelling via a similar track or close in space/time, which their method does. However, this appears to by default just largely characterise the main storm tracks of the globe (Figs. 2,4). The standard view of clustering (e.g. Mailier et al., Pinto et al., Priestley et al.) characterises clustering as an abnormal rate of cyclone occurrences. Therefore, I would like to see more justification from the authors on their choice of thresholds for their detection method. If they are more strict, what events do they identify? Do signals become weaker as the frequency of events decreases, or are a different subset of events identified. Please clarify this and consider adding new results into the manuscript.

**Minor Comments**
1. L17/18 – rephrase to "is often quantified to be associated with European weather extremes"
2. L23 – not all references discussed in L22 are related to statistical quantification of clustering
3. L27-29 – I find some of your discussion of overdispersive hard to follow here. The sentence "In contrast, a region is overdispersive when cyclones occur less regularly compared to a Poisson process." Is to me incorrect. Overdispersive is the deviation from a poisson process. Perhaps stated as when the rate of cyclones is variable compared to a Poisson process?
4. L32 "generally small and have large uncertainties"
5. L33 – how is it a problem to define clustering in a relative sense?
6. L41 – I would argue that these studies do not use an "impact-based definition", but instead the clustering method introduced by Pinto et al. (2014) classifies storms into clusters that then happen to cause impacts.

7. L48/49 – this is incorrect. The algorithm does not a priori assume clustering is due to secondary cyclogenesis. Just that secondary cyclogenesis often contributes to clustering.

8. L78 – why are you using ERA-Interim and not ERA5. Interim is now very outdated and limited in time.

9. L89 – 'meters'

10. L104 – I am confused as to your overlapping criteria. On L94 you mention a 36 hour threshold, which I believe is the time difference for a cyclone to be within 1.5 Rossby radius, then what is the 2 days relating to? Must they be within 36 hours/1.5RR for 2 days of each cyclones lifecycle? This whole section is quite hard to follow so I suggest editing to improve readability.

11. L114 – 'yields all cyclone clusters' – what does this mean?

12. L115-120 – for analysis I understand that you only take the part of the track that contributes to that part ofa cluster in the analysis. Does this mean that in the track densities and intensity calculations, you only use fractional parts of the tracks? Please clarify this? If later on you search for the most intense storm in a cluster, does this mean you are not using all the information of each track to do this analysis?

13. L121 – stagnant clusters do not travel far, but in your schematic of figure 1 track 2 does travel a long way. If you are taking just the end part of the track at 'stagnant' you can't really say that it has not travelled far in my opinion.

14. L129 – the statement on only using the connected parts of tracks in clustered cyclones, does this impact your findings?

15. L139 – how 'similar'? Please give some more quantitative information to this statement

16. L143 and figure 2 – I don't understand the units here or how to interpret them. Is this the fraction of total cyclones, and then Fig. 2c,f is the fraction of clustered compared to clustered+solo? Please explain these units and the interpretation of the figure more clearly in the caption and the text.

17. L147-149 – these irregularities are surely the interesting part, as your method largely detects regular activity. Can you detect such irregularities using this method?

18. L151 I would argue from Figure 2a that solo cyclones are not just on the exit of the storm track, but mainly where cyclones just have more infrequent occurrence. Consider rephrasing.

19. L156 – you are comparing different things here. In Priestley et al. (2020) this percentage is of family cyclones in total, these do not have to contribute to clusters as in this analysis here.

20. Figure 2 – I would suggest making the upper limit of your colourbar higher as it is hard to detect some of the maxima within your figures due to the colour saturation. This is especially the case in b/f.

21. L197/198 – some reference editing is needed here, should be in brackets.

22. L209/210 – would you not expect the strongest cyclone in a cluster to be stronger than random most of the time anyway, as you are preselecting a strong cyclone?

23. L225/226 – to clarify this, for this analysis you calculate the $90^{th}$ percentile at all locations and this is how often a cyclone has intensity exceeding this value?

24. Figure 8 – caption should be blue shading for your stagnant clusters

25. Figure 10 – I don't fully understand what you are using to generate this information. Is the length of overlap for how long the cyclones overlap for from the point of

genesis? The same with Time of overlap, is this for two connected storms, or just the length of the cluster? Please make the text associated with this figure clearer as to how this is interpreted and generated.

26. Figure 11 – are these results the same if you use something like MSLP or cyclone size? Theories on clusters are that the final storm is more intense and larger and so would be good to document alongside this result.

---

## Author Comment (AC1)

**Public respond to reviewers egusphere-2024-3404**

The authors thank the editor and the reviewers for their constructive comments and suggestions. Please, see below our public responses.

**Response to the reviewers**

**Reviewer 1**

**Reviewer Comment 1.1** — This work introduces a new climatology of cyclone clustering using a novel detection and classification algorithm. The work and associated findings are interesting, and some new insights are provided, however I question the definition of clustering used here compared to some of the previous scientific literature. The authors use a fixed clustering threshold everywhere and largely identify clustering in the core of the storm track, where more cyclones are found. Whereas previous efforts have identified clustering as abnormal periods of high cyclone activity. Therefore, I would like to see more justification from the authors as to their methodological choices and explanation of novelties relative to prior studies, and how this work differs from a simple classification of the storm tracks. I recommend major revisions for this work, and detail my points, both major and minor, below.

**Reply**: Thank you for your constructive comments. It is correct that several previous definitions define cyclone clustering as abnormal periods of high cyclone activity, either through statistical measures or a running-mean occurrence threshold for cyclones in a geographically confined domain. However, at the same time, most of these efforts also refer back to the original work by Bjerknes and Solberg (1919) on cyclone families, which is based on the space-time proximity of cyclone tracks in a not a priori confined domain. Our detection algorithm tries to be as true as possible to this original definition of cyclone families, requiring a space-time proximity of cyclone tracks. Of course, the space-time proximity is also implicit in some of the other previously introduced cyclone cluster detection algorithms, though they either constrain the analysis to a pre-defined geographic location or use cyclone track densities for statistical assessments. The former is limited to a pre-defined domain, whereas the latter cannot be used to directly identify the cyclone tracks that are part of a specific cluster.
The motivation for our approach was to develop an algorithm that can be applied globally, while concomitantly retaining the information about the specific cyclone tracks are part of a respective cluster. With this information at hand, new questions about cyclone clustering can be addressed, such as: What are preferred regions for cyclone clustering? Do cyclones cluster more locally in time without necessarily moving much, or do they cluster in space and time, i.e., they move along similar paths? Are there differences in the characteristics of the different types of clusters? Are there structural differences in clustered and non-clustered cyclones? We believe that these are highly relevant questions to further characterise the occurrence of cyclone clusters and to assess dynamical differences between clustered and non-clustered cyclones. We will make these points clearer in the revised version of the manuscript.

**Reviewer Comment 1.2** — My main concern surrounds the choice of the algorithm and justifications made by the authors. The method is to group cyclone travelling via a similar track or

close in space/time, which their method does. However, this appears to by default just largely characterise the main storm tracks of the globe (Figs. 2,4). The standard view of clustering (e.g. Mailier et al., Pinto et al., Priestley et al.) characterises clustering as an abnormal rate of cyclone occurrences. Therefore, I would like to see more justification from the authors on their choice of thresholds for their detection method. If they are more strict, what events do they identify? Do signals become weaker as the frequency of events decreases, or are a different subset of events identified. Please clarify this and consider adding new results into the manuscript.

**Reply**: Thank you for your constructive comments on the justification of our method. It is correct that the cyclone clusters detected by our algorithm have a relatively high occurrence in the storm tracks region. This is not unexpected, as a higher occurrence of cyclones in general would also increase the liklihood of space-time proximity demanded by our algorithm. However, we also clearly show that only a fraction of maximum around 50-60 percent of cyclones are clustered in these storm track regions. This implies that not all cyclones must be characterised by a space-time proximity. Hence, despite the "normal-ness" of the storm track, the occurrence of cyclone clusters is not the norm in these regions either.

Furthermore, it is not quite correct to state that previous methods only characterised clustering as an abnormal rate of cyclone occurences. The method of Priestley et al. (2016), for example, uses a threshold of having at least 4 cyclones in a 7-day running of cyclone occurrence to detect cyclone clusters, which gives similar results in the storm tracks region if applied globally (see top left panel Fig. 1), even when only using the more intense cyclones (see top right panel Fig. 1). When only retaining the most extreme cyclones, the signal shifts towards the end of the stormtrack (see lower panel Fig. 1). Hence, one could argue that the reason that cyclone clustering is abnormal in the Priestley et al. algorithm is due to the choice of a specific region as well as cyclone intensity. Also the Mailier et al. (2006) method does not directly characterise abnormal rates of cyclone occurence, but rather the abnormal variability of cyclone occurrence.

Originally, Bjerknes and Solberg (2022) described cyclone families as the common evolution in the North-Atlantic storm track. However, even when adopting a cyclone clustering detection in their spirit, we find that only a fraction of cyclones actually clusters in the storm track region, with diminishing numbers towards the storm track exit, highlighting the abnormality of cyclone clustering in these regions. Most importantly, our diagnostic is motivated by enabling a dynamical perspective on cyclone clustering to investigate the mechanisms behind cyclone clustering, without a priori limiting the view to selected regions or the most intense cyclones. Hence, there was a need for a global detection scheme that applies a space-time proximity criteria to define cyclone clustering. It is this need that our detection algorithm replies to. We will make this clearer in a revised version of the manuscript.

We also take the reviewer's point on presenting a more complete discussion on the sensitivity of our algorithm to the chosen thresholds. We have, of course, tested a wide range of threshold and will include a more thorough discussion of the sensitivities in a revised version of the manuscript.

**Reviewer Comment 1.3** — L17/18 – rephrase to "is often quantified to be associated with European weather extremes"

**Reply**: We will rephrase accordingly.

**Reviewer Comment 1.4** — L23 – not all references discussed in L22 are related to statistical quantification of clustering

[Figure]

Figure 1: Cyclone clustering climatology for DJF in the Northern Hemisphere using a threshold of (upper left) at least 4 cyclones in a 7-day running mean in a local 700 km radius. Plotted in shading is the percentage of the time steps that this condition is satisfied. (upper right) The same as upper left, but only using intense cyclones (above a Laplacian of 2.0). (lower panel) The same as upper left, but for 3 cyclones in a 7-day running mean and only using the cyclones with an intensity above the local 5.0% pressure DJF quantile.

**Reply**:  We will carefully check these references and adjust them accordingly.

**Reviewer Comment 1.5**  —  L27-29 – I find some of your discussion of overdispersive hard to follow here. The sentence "In contrast, a region is overdispersive when cyclones occur less regularly compared to a Poisson process." Is to me incorrect. Overdispersive is the deviation from a poisson process. Perhaps stated as when the rate of cyclones is variable compared to a Poisson process?

**Reply**:  It is correct that over-dispersion is defined as the presence of greater variability (statistical dispersion) than expected based on a statistical model. However, under-Our usage of the wording "regular" stems from the way Mailier et al used this term when referring to their results of over- and under-dispersion. Given the potential confusion about these statements, we will clarify these points in a revised version of the manuscript.

**Reviewer Comment 1.6**  —  L32 "generally small and have large uncertainties"

**Reply**:  We will rephrase accordingly.

**Reviewer Comment 1.7**  —  L33 – how is it a problem to define clustering in a relative sense?

**Reply**:  It is not a problem in itself, but using a relative measure has the disadvantages outlined in the sentence directly following in line 34. We will try to connect these sentences better to streamline the logic. In addition, we will make it clearer that our aim is to be able to directly identify the specific cyclones that are part of a cluster, which will allow for a more dynamical analysis and comparison of clustered cyclones.

**Reviewer Comment 1.8**  —  L41 – I would argue that these studies do not use an "impact-based definition", but instead the clustering method introduced by Pinto et al. (2014) classifies storms into clusters that then happen to cause impacts.

**Reply**:  We referred to them as impact-based definitions, because these clustering detections pre-select only the most intense cyclones and a priori focus on a specific geographic region. However, we see the reviewer's point and will try to rephrase this formulation.

**Reviewer Comment 1.9**  —  L48/49 – this is incorrect. The algorithm does not a priori assume clustering is due to secondary cyclogenesis. Just that secondary cyclogenesis often contributes to clustering.

**Reply**:  Thank you for this clarification. We will adjust the text accordingly.

**Reviewer Comment 1.10**  —  L78 – why are you using ERA-Interim and not ERA5. Interim is now very outdated and limited in time.

**Reply**:  Our analysis was already performed before a more complete ERA5 dataset was available. Preliminary testing indicates that our results are not sensitive to the choice of reanalysis. We will comment on that in a revised version of the manuscript and perform further tests.

**Reviewer Comment 1.11**  —  L89 – 'meters'

**Reply**: We will correct accordingly.

**Reviewer Comment 1.12** — L104 – I am confused as to your overlapping criteria. On L94 you mention a 36 hour threshold, which I believe is the time difference for a cyclone to be within 1.5 Rossby radius, then what is the 2 days relating to? Must they be within 36 hours/1.5RR for 2 days of each cyclones lifecycle? This whole section is quite hard to follow so I suggest editing to improve readability.

**Reply**: Thank you for pointing out this confusion. We agree that the description of the methodology would benefit from further clarification. We will further clarify the steps in our clustering algorithm and will adjust the text accordingly.

**Reviewer Comment 1.13** — L114 – 'yields all cyclone clusters' – what does this mean?

**Reply**: This means that we 'obtain all cyclone clusters'. We will rephrase for better readability.

**Reviewer Comment 1.14** — L115-120 – for analysis I understand that you only take the part of the track that contributes to that part of a cluster in the analysis. Does this mean that in the track densities and intensity calculations, you only use fractional parts of the tracks? Please clarify this? If later on you search for the most intense storm in a cluster, does this mean you are not using all the information of each track to do this analysis?

**Reply**: Thank you for pointing out this potential point of confusion. We indeed only use the fractional parts of the tracks for the track density analysis. For cyclone intensity, however, we use the entire track. We see that this is slightly inconsistent and will rectify this in the revised version of the manuscript.

**Reviewer Comment 1.15** — L121 – stagnant clusters do not travel far, but in your schematic of figure 1 track 2 does travel a long way. If you are taking just the end part of the track at 'stagnant' you can't really say that it has not travelled far in my opinion.

**Reply**: Thank you for pointing this out. The reviewer's interpretation is correct. While the clustered part of the cyclone tracks is usually geographically very confined, individual tracks of these cyclones can cover a larger geographic region. We will clarify and adapt this in the revised version of the manuscript.

**Reviewer Comment 1.16** — L129 – the statement on only using the connected parts of tracks in clustered cyclones, does this impact your findings?

**Reply**: When using the whole track, we obtain a qualitatively similar distribution of cyclone cluster density for the Northern Hemisphere winter season (see Fig. 2). The main difference is that the absolute density is higher when considering the entire track. We will comment on this finding in the revised version of the manuscript.

**Reviewer Comment 1.17** — L139 – how 'similar'? Please give some more quantitative information to this statement

[Figure]

Figure 2: Cyclone clustering climatology for DJF in the Northern Hemisphere using the entire track instead of only 'connected' parts. Shading denotes fraction of times a clustered cyclone is at a given location in a 1000 km$^2$ area.

**Reply**: We will quantify this statement in more detail and comment on it in the revised version of the manuscript.

**Reviewer Comment 1.18** — L143 and figure 2 – I don't understand the units here or how to interpret them. Is this the fraction of total cyclones, and then Fig. 2c,f is the fraction of clustered compared to clustered+solo? Please explain these units and the interpretation of the figure more clearly in the caption and the text.

**Reply**: Thank you for pointing out this potential confusion. Fractions are indeed with respect to total cyclones. We will clarify this in the caption accordingly.

**Reviewer Comment 1.19** — L147-149 – these irregularities are surely the interesting part, as your method largely detects regular activity. Can you detect such irregularities using this method?

**Reply**: While we appreciate the interest of the reviewer in this irregular part alluded to in previous publications, we would argue that this interest is subjective. Here, we identify cyclone clusters based on a space-time proximity, without a priori demanding a statistical focus on irregularities. We detect cyclone clusters both in regions where they occur as irregularities as well as in regions where they are more common. Being able to distinguish these regions is valuable, as it allows for the further characterisation of cyclone clusters as well as focus on their dynamical differences.

**Reviewer Comment 1.20** — L151 I would argue from Figure 2a that solo cyclones are not just on the exit of the storm track, but mainly where cyclones just have more infrequent occurrence. Consider rephrasing.

**Reply**: The reviewer has a point. We will rephrase accordingly.

**Reviewer Comment 1.21** — L156 – you are comparing different things here. In Priestley et al. (2020) this percentage is of family cyclones in total, these do not have to contribute to clusters as in this analysis here.

**Reply**: Thank you for pointing this out. We will adjust the text accordingly.

**Reviewer Comment 1.22** — Figure 2 – I would suggest making the upper limit of your colourbar higher as it is hard to detect some of the maxima within your figures due to the colour saturation. This is especially the case in b/f.

**Reply**: Thank you for the suggestion. We will adjust the figure accordingly.

**Reviewer Comment 1.23** — L197/198 – some reference editing is needed here, should be in brackets.

**Reply**: Thank you. We will check and correct accordingly.

**Reviewer Comment 1.24** — L209/210 – would you not expect the strongest cyclone in a cluster to be stronger than random most of the time anyway, as you are preselecting a strong cyclone?

**Reply**: In contrast to previous cyclone cluster detection algorithms, we do not pre-select strong cyclones, i.e., we do not employ an intensity threshold for cyclones considered in our clustering algorithm. Hence, in contrast to previous studies, we can perform this distinction and present it in our manuscript. We will make sure to clarify and emphasise this in the revised version of the manuscript.

**Reviewer Comment 1.25** — L225/226 – to clarify this, for this analysis you calculate the 90th percentile at all locations and this is how often a cyclone has intensity exceeding this value?

**Reply**: Yes, the interpretation of the reviewer is correct. We will clarify this in a revised version of the manuscript accordingly.

**Reviewer Comment 1.26** — Figure 8 – caption should be blue shading for your stagnant clusters

**Reply**: Thank you. We will fix this typo accordingly.

**Reviewer Comment 1.27** — Figure 10 – I don't fully understand what you are using to generate this information. Is the length of overlap for how long the cyclones overlap or from the point of genesis? The same with Time of overlap, is this for two connected storms, or just the length of the cluster? Please make the text associated with this figure clearer as to how this is interpreted and generated.

**Reply**: Thank you for pointing out this confusion. These overlaps are defined in the methods section, though the manuscript obviously would benefit from referring the reader back to the methods section when discussing these results to avoid confusion. We will clarify and rephrase accordingly.

**Reviewer Comment 1.28** — Figure 11 – are these results the same if you use something like MSLP or cyclone size? Theories on clusters are that the final storm is more intense and larger and so would be good to document alongside this result.

**Reply**: We did in fact perform this analysis also using MSLP and the results are very similar. We will refer to these results in the revised version of the manuscript. Regarding the size of cyclones. There is not generally accepted definition of the size of a cyclone, as it could be based on vorticity, MSLP, or other intensity or structure measures. We hence refrain from including such a discussion. We do not find evidence that the final storm is more intense and will make sure to clarify this in the revised versin of the manuscript.